# Histone H3K9 Demethylase JMJD2B Plays a Role in LXRα-Dependent Lipogenesis

**DOI:** 10.3390/ijms21218313

**Published:** 2020-11-05

**Authors:** Ji-Hyun Kim, Dae Young Jung, Hye-Ran Kim, Myeong Ho Jung

**Affiliations:** Division of Longevity and Biofunctional Medicine, School of Korean Medicine, Pusan National University, Yangsan 50612, Korea; kimji77@pusan.ac.kr (J.-H.K.); dyjung999@naver.com (D.Y.J.); hrkim308@gmail.com (H.-R.K.)

**Keywords:** hepatic steatosis, histone demethylase Jumonji domain-containing protein 2B, histone methylation, ligand-activated liver X receptor α, lipogenesis

## Abstract

Ligand-activated liver X receptor α (LXRα) upregulates the expression of hepatic lipogenic genes, which leads to triglyceride (TG) accumulation, resulting in nonalcoholic fatty liver disease (NAFLD). Thus, LXRα regulation may provide a novel therapeutic target against NAFLD. However, histone methylation-mediated epigenetic regulation involved in LXRα-dependent lipogenesis is poorly understood. In this study, we investigated the functional role of the histone demethylase Jumonji domain-containing protein 2B (JMJD2B) in LXRα-dependent lipogenesis. JMJD2B expression level was upregulated in HepG2 cells treated with LXRα agonist T0901317 or palmitate and the liver of mice administered with T0901317 or fed a high-fat diet. Knockdown of JMJD2B using siRNA abrogated T0901317-induced LXRα-dependent lipogenic gene expression and lowered intracellular TG accumulation. Conversely, overexpression of JMJD2B in HepG2 cells upregulated the expression of LXRα-dependent lipogenic genes, in line with increased intracellular TG levels. JMJD2B overexpression or T0901317 treatment induced the recruitment of JMJD2B and LXRα to LXR response elements (LXRE) in the promoter region of LXRα-target gene and reduced the enrichment of H3K9me2 and H3K9me3 in the vicinity of the LXRE. Furthermore, JMJD2B enhanced T0901317 or LXRα-induced transcriptional activities of reporters containing LXRE. A co-immunoprecipitation assay revealed that JMJD2B interacted with activated LXRα. Moreover, overexpression of JMJD2B in mice resulted in upregulation of hepatic LXRα-dependent lipogenic genes, consistent with development of hepatic steatosis. Taken together, these results indicate that JMJD2B plays a role in LXRα-mediated lipogenesis via removing the repressive histone marks, H3K9me2 and H3K9me3, at LXRE, which might contribute to hepatic steatosis.

## 1. Introduction

Nonalcoholic fatty liver disease (NAFLD), an etiology of chronic liver disease, involves hepatic steatosis, steatohepatitis, and cirrhosis and is associated with metabolic disorders including insulin resistance, type 2 diabetes, and obesity [1]. Hepatic steatosis, a hallmark of NAFLD, is caused by an imbalance in triglyceride (TG) synthesis, including de novo lipogenesis (DNL), the uptake of free fatty acids from adipose tissue or diet, and TG removal via fatty acid oxidation, very low-density lipoprotein (VLDL) secretion, and lipophagy [2]. Thus, excess DNL can contribute to the development of NAFLD through the overproduction of fatty acids [2]. DNL is stimulated by lipogenic transcription factors including sterol regulatory element-binding protein-1c (SREBP-1c; gene *SREBF1*), carbohydrate responsive element binding protein (ChREBP), and liver X receptor (LXR; gene *NR1H3*), which positively regulate the expression of lipogenic genes including fatty acid synthase (*FAS*), acetyl CoA carboxylase (*ACC*), and stearoyl-CoA desaturase 1 (*SCD1*). Therefore, the regulation of DNL is a potential therapeutic method for NAFLD. Recently, studies on the role of epigenetics in the regulation of hepatic lipogenesis have been performed [3]. However, epigenetic regulation of lipogenesis through histone methylation is poorly understood.

LXRs are ligand-activated transcription factors that play an important role in promoting lipogenesis [4]. There are two subtypes of LXRs: LXRα and LXRβ. LXRα is abundantly expressed in the liver, adipose tissue, and kidney, whereas LXRβ is expressed ubiquitously. Ligand-activated LXRs form obligate heterodimers with retinoid X receptors (RXRs) and regulate expression of target genes containing LXR response elements (LXRE). LXRα directly stimulates the expression of LXRα-dependent lipogenic genes, including *FAS*, *ACC*, and *SCD1*, through binding to LXRE in their promoter regions. In addition, LXRα upregulates the expression of SREBP-1c, a critical regulator of lipogenesis, through binding to the LXRE in the promoter region, thereby promoting the expression of its downstream lipogenic genes, including *FAS, ACC*, and *SCD1* [4]. Thus, the LXRα-induced activation of SREBP1c plays an important role in promoting hepatic lipogenesis. Accordingly, activation of LXRα increases hepatic TG accumulation and results in hepatic steatosis [5]. Although recent studies have revealed several coactivators of LXRα [4], histone modifiers involved in the epigenetic regulation of LXRα-dependent lipogenesis are not well understood.

Histone methylation marks are responsible for the epigenetic regulation of chromatin structure through addition or removal of methyl groups from lysine residues of histone tails [6]. Mono-, di-, and tri-methylation marks and their respective demethylation of lysine residues within histones H3 and H4 act as epigenetic switches that can either activate or repress transcription. Histone H3 tri-methylation at lysine 4 (H3K4me3) and lysine 36 (H3K36me3) activate transcription, whereas tri-methylation or di-methylation at lysine 9 (H3K9me3 or H3K9me2) and tri-methylation at lysine 27 (H3K27me3) generally repress transcription [6]. Histone demethylases remove methyl groups from modified histones, thereby activating or repressing gene transcription. The histone demethylase Jumonji domain-containing protein 2B (JMJD2B, also known as KDM4B) is a Jumonji (Jmj)—containing histone demethylase that removes repressive di- and tri-methylation marks at lysine 9 (H3K9me2/3) on histone H3, converting the marks to the monomethylated state; thus, JMJD2B functions as a transcription activator [7,8]. It has been reported that JMJD2B regulates cellular responses via cell differentiation [9], cell proliferation [10], DNA damage [11], steroid hormone [12], and energy status [13]. Notably, JMJD2B expression is enhanced in various cancers and plays an important role in the promotion of tumorigenesis [14].

Previously, we demonstrated the functional role of JMJD2B in hepatic steatosis [15]. JMJD2B upregulated the expression of peroxisome proliferator-activated receptor γ2 (*PPAR*γ2) and its steatosis target genes including *CD36* and fatty acid-binding protein (*FABP*), which are necessary for fatty acid uptake, thus resulting in the development of hepatic steatosis. Meanwhile, we also observed that JMJD2B expression increased in HepG2 cells and mice treated with the LXRα agonist T0901317, coinciding with the upregulated expression of LXRα-dependent lipogenic genes, suggesting that JMJD2B is involved in LXRα-dependent lipogenesis. Therefore, in the current study, we investigated whether JMJD2B plays a role in the LXRα-mediated stimulation of lipogenesis, besides the promotion of fatty acid uptake via upregulation of PPARγ2 expression. Here, we provide evidence that JMJD2B is an epigenetic co-activator of LXRα, and functions to stimulate LXRα-dependent lipogenesis.

## 2. Results

### 2.1. JMJD2D Expression Was Upregulated in HepG2 Cells and the Livers of Mice Treated with LXRα Agonist

To investigate whether JMJD2D is correlated with LXRα-activated lipogenesis, we examined JMJD2B expression in LXRα agonist T0901317-treated HepG2 cells. Treatment with T0901317 elevated intracellular TG levels in HepG2 cells (Figure 1A). Consistent with elevated TG levels, qPCR revealed that *JMJD2B* mRNA expression was upregulated to higher levels in T0901317-treated HepG2 cells than in untreated cells (Figure 1B), which was consistent with increased mRNA expression levels of *NR1H3* (Figure 1C), *SREBF1* (Figure 1D), and LXRα or SREBP1c target lipogenic genes, *FAS*, *ACC*, and *SCD1* (Figure 1E). Western blotting also revealed that the protein level of JMJD2B was increased in T0901317-treated HepG2 cells, concomitant with increased protein expression of LXRα, SREBP1c, and FAS (Figure 1F). We further assessed JMJD2D expression in the liver of mice administrated with T0901317 for 5 days. T0901317 administration converted a healthy liver to a pale, steatotic liver (Figure 1G). Further, the liver weight (Figure 1H) of the T0901317-treated mice was higher than that of the non-treated mice. Consistent with steatotic liver, western blotting revealed that T0901317 administration also enhanced JMJD2B protein expression, concomitant with increased levels of LXRα and LXRα-dependent lipogenic proteins, SREBP1c, and FAS (Figure 1I).

Furthermore, we evaluated the mRNA and protein levels of JMJD2B in palmitate (PA)-treated HepG2 cells. PA was used to make an in vitro steatotic cell model because PA induces the expression of lipogenesis genes, resulting in TG synthesis. Treatment with PA increased intracellular TG levels (Figure 2A) and mRNA levels of *JMJD2B* (Figure 2B), *NR1H3* (Figure 2C), *SREBF1* (Figure 2D), and LXRα target lipogenic genes, including *FAS, ACC*, and *SCD1* (Figure 2E). Western blotting also revealed that the protein level of JMJD2B was increased in PA-treated HepG2 cells, concomitant with increased LXRα, SREBP1c, and FAS protein expression levels (Figure 2F). Additionally, we assessed JMJD2D expression in the liver of mice fed a high-fat diet (HFD). HFD resulted in a pale, steatotic liver (Figure 2G); HFD-fed mice exhibited higher liver weight (Figure 2H) than mice fed a normal diet (ND). Consistent with steatotic liver, western blotting revealed that HFD enhanced the protein expression of JMJD2B, together with increased expression levels of LXRα and LXRα-dependent lipogenic proteins, including SREBP1c and FAS (Figure 2I). Taken together, these results indicate that JMJD2B is associated with LXRα-activated lipogenesis.

### 2.2. Knockdown of JMJD2D Prevented LXRα-Dependent Lipogenesis

To determine the functional role of JMJD2B in LXRα-dependent lipogenesis, we investigated whether JMJD2D knockdown affects LXRα-dependent gene expression and TG accumulation in HepG2 cells. To knockdown JMJD2B, HepG2 cells were transfected with JMJD2B siRNA. Both qPCR and western blotting revealed that T0901317 treatment increased JMJD2B mRNA and protein levels in HepG2 cells, which were significantly reduced by siRNA treatment (Figure 3A). Then, we measured intracellular TG levels and the expression of LXRα-dependent lipogenic genes in JMJD2B siRNA-transfected HepG2 cells. As shown in Figure 3B, knockdown of JMJD2B significantly reduced T0901317-induced intracellular TG levels in HepG2 cells. Consistent with the decreased levels of TG, JMJD2B knockdown prevented the T0901317-induced mRNA expression of LXRα-dependent lipogenic genes including *SREBF1*, *FAS*, *ACC*, and *SCD1* (Figure 3C). Western blotting also revealed that knockdown of JMJD2B reduced the T0901317-induced increase in protein levels of LXRα and LXRα-dependent lipogenic genes in HepG2 cells (Figure 3D). These results indicate that JMJD2B plays a role in LXRα-dependent lipogenesis and TG accumulation.

### 2.3. Overexpression of JMJD2B Induced LXRα-Dependent Lipogenesis in HepG2 Cells

We next performed a gain-of-function study to determine whether JMJD2B directly increases lipogenesis and intracellular TG accumulation in HepG2 cells. To overexpress JMJD2B ectopically, HepG2 cells were infected with an adenovirus containing JMJD2B (Ad-JMJD2B). Both qPCR and western blotting revealed that the mRNA and protein expression of JMJD2B was significantly elevated to higher levels in Ad-JMJD2B-infected HepG2 cells than in Ad-GFP-infected HepG2 cells (Figure 4A).

Consistent with the upregulation of JMJD2B, adenovirus-mediated overexpression of JMJD2B led to increased intracellular TG levels in HepG2 cells (Figure 4B). Additionally, we assessed the expression of LXRα and LXRα-dependent lipogenic genes in Ad-JMJD2B-infected HepG2 cells. Western blotting showed that overexpression of JMJD2B resulted in increased protein levels of LXRα, SREBP1c, and FAS (Figure 4C). In addition, qPCR analysis revealed that overexpression of JMJD2B increased the mRNA expression levels of *NR1H3* and LXRα-dependent lipogenic genes *SREBF1*, *FAS, ACC*, and *SCD1* (Figure 4D). Taken together, these results indicate that JMJD2B enhances the expression of LXRα and LXRα-dependent lipogenic genes, resulting in hepatic steatosis.

### 2.4. Overexpression of JMJD2B Reduced the Enrichment of H3K9me2 and H3K9me3 at LXREs on the SREBF1 Promoter via the Recruitment of JMJD2B and LXRα

We then explored the possible mechanism by which JMJD2B stimulates LXRα-mediated lipogenic genes. LXRα enhances the expression of lipogenic genes including *SREBF1*, *FAS*, *ACC*, and *SCD1*, through binding to LXREs on their promoters [4]. JMJD2B is a histone demethylase that removes H3K9me2/me3 marks in methylated histones [8]. Thus, we hypothesized that JMJD2B removes both H3K9me2 and H3K9me3 from LXREs on the promoter regions of LXRα-dependent lipogenic genes, thus upregulating the expression of the lipogenic genes. To test this hypothesis, we first assessed the recruitment of JMJD2B and LXRα to LXREs on the promoter of the LXRα-target gene *SREBF1* in JMJD2B-overexpressed HepG2 cells. The chromatin of HepG2 cells infected with Ad-JMJD2B was immunoprecipitated with antibody against JMJD2B or LXRα, and the immunoprecipitated chromatin was subjected to qPCR using primers spanning the LXRE on the *SREBF1* promoter. As shown in Figure 5A,B, chromatin immunoprecipitation (ChIP)-qPCR revealed that overexpression of JMJD2B significantly increased the enrichment of JMJD2B and LXRα at the LXRE of the *SREBF1* promoter; this suggests that JMJD2B is recruited to the LXREs of LXRα-dependent lipogenic genes to activate them. We then analyzed the enrichment of H3K9me2 and H3K9me3 in the LXRE of the *SREBF1* promoter in JMJD2B-overexpressed HepG2 cells using antibodies against H3K9me2 and H3K9me3. ChIP-qPCR revealed a decrease in H3K9me2 and H3K9me3 enrichment in Ad-JMJD2B-infected HepG2 cells (Figure 5C,D), indicating that JMJD2B is recruited to the LXRE region and thus removes the repressive histone marks H3K9me2 and H3K9me3, resulting in the stimulation of LXRα-dependent lipogenic genes.

### 2.5. LXRα Activation Increased JMJD2B Recruitment to LXREs and Reduced H3K9me2 and H3K9me3 Enrichment in the SREBF1 Promoter

Next, we attempted to determine whether LXRα activation affects the enrichment of H3K9me2 and H3K9me3 on the *SREBF1* promoter via recruitment of JMJD2B to the LXRE region. ChIP-PCR was performed in T0901317-treated HepG2 cells using antibodies against JMJD2B, LXRα, H3K9me2, and H3K9me3, and primers spanning the LXRE on the *SREBF1* promoter. ChIP-PCR revealed that T0901317 treatment increased the recruitment of JMJD2B (Figure 6A) as well as LXRα (Figure 6B) to the LXRE of the *SREBF1* promoter. Consistent with increased recruitment of JMJD2B, T0901317 treatment reduced the enrichment of H3K9me2 (Figure 6C) and H3K9me3 (Figure 6D) at the LXRE of the *SREBF1* promoter.

### 2.6. JMJD2B Stimulated LXRα-Activated Transcriptional Activity

We further investigated whether JMJD2B affected the LXRα-mediated transcriptional activation of lipogenic genes. To this end, we performed luciferase reporter assays using three luciferase reporters including 3xLXRE-Luc containing 3 copies of LXRE, *SREBF1*-Luc which has a *SREBF1* promoter containing LXRE, and *FAS*-Luc which has a *FAS* promoter containing LXRE in HepG2 cells. As shown in Figure 7A, treatment with T0901317 stimulated the promoter activities of three reporters, including 3xLXRE-Luc, *SREBF1*-Luc, and *FAS*-LXRE, which were more enhanced by JMJD2B overexpression. Additionally, LXRα overexpression stimulated the transcriptional activities of three reporters, which were also elevated by JMJD2B overexpression (Figure 7B). Taken together, these results suggest that JMJD2B enhances LXRα-mediated transcriptional activity.

### 2.7. JMJD2B Interacted Directly with Activated LXRα

Finally, we speculated that JMJD2B might be recruited to the LXRE region through a direct interaction with LXRα as it is bound to LXRE. Thus, to determine whether JMJD2B interacts with LXRα, we performed a co-immunoprecipitation assay in HepG2 cells infected with adenovirus expressing JMJD2B (Ad-JMJD2B) and then incubated in the presence or absence of T0901317. As shown in Figure 8, LXRα could be pulled down from JMJD2B antibody-mediated immunoprecipitants in only both Ad-JMJD2B-infected and T0901317-treated HepG2 cells. We also conducted a co-immunoprecipitation assay, in which LXRα was first immunoprecipitated and followed by immunoblotting with an antibody against JMJD2B. We found that immunoprecipitation of LXRα with LXRα antibody pulled down JMJD2B protein in only both Ad-JMJD2B-infected and T0901317-treated HepG2 cells (Appendix A). These results suggest that JMJD2B can interact with activated LXRα. Taken together, these results indicate that JMJD2B might be recruited to LXREs of LXRα-dependent lipogenic genes by interaction with activated LXRα.

### 2.8. Adenovirus-Mediated JMJD2B Overexpression Stimulated LXRα-Dependent Lipogenesis and Induced Hepatic Steatosis in Mice

To further substantiate the functional role of JMJD2B in LXRα-dependent lipogenesis in vivo, we examined the expression of LXRα-dependent lipogenic genes in the liver of JMJD2B-overexpression mice. We injected recombinant Ad-JMJD2B into the tail vein of C57BL/6J mice to overexpress hepatic JMJD2B. qPCR revealed that *JMJD2B* mRNA was successfully overexpressed in the livers of Ad-JMJD2B-injected mice (Figure 9A). Consistent with the increased mRNA levels of *JMJD2B*, mRNA levels of *NR1H3* (Figure 9B) and LXRα-dependent lipogenic genes *SREBF1*, *FAS*, *ACC*, and *SCD1* (Figure 9C) were increased in the livers of Ad-JMJD2D-injected mice. We then investigated the effect of JMJD2B on the development of hepatic steatosis in Ad-JMJD-injected mice. Liver histology showed that adenovirus-mediated overexpression of JMJD2B converted a healthy liver into a yellow-colored liver (Figure 9D). Further, hepatic TG levels were significantly higher in Ad-JMJD2B-injected mice than in Ad-GFP-injected mice (Figure 9E). Additionally, hematoxylin and eosin (H&E) and oil red O (ORO) staining revealed that Ad-JMJD2B-injected mice had more abundant lipid droplets than Ad-GFP-injected mice (Figure 9F). Taken together, these results indicate that in vivo JMJD2B overexpression stimulates LXRα-dependent lipogenesis, resulting in the development of hepatic steatosis in HFD-fed mice.

## 3. Discussion

Previously, we reported that JMJD2B enhances PPARγ2 expression by removing the repressive histone marks H3K9me2 and H3K9me3 in the promoter of PPARγ2, stimulating the expression of *PPARγ2* and its steatosis target genes such as *CD36* and *FABP*, resulting in the development of hepatic steatosis [15]. In the current study, we speculated that JMJD2B might also play an epigenetic role in LXRα-activated lipogenesis, leading to the development of NAFLD. Our current data demonstrate that JMJD2B induces LXRα-dependent lipogenesis by removing repressive histone marks H3K9me2 and H3K9me3 near LXREs of lipogenic gene promoters. These results suggest that JMJD2B is an epigenetic coactivator of LXRα to stimulate LXRα-dependent lipogenic genes.

The earliest stage of NAFLD is hepatic steatosis, which reflects TG accumulation in hepatocytes as a result of increased de novo lipogenesis and increased uptake of free fatty acids that exceeds the rate of fatty acid oxidation, VLDL secretion, and lipophagy [2]. Recently, several studies have reported a link between the regulation of hepatic steatosis and histone modifications [15,16,17,18]. Histone deacetylase 3 (HDAC3) is an epigenetic regulator associated with hepatic steatosis [16,17]. HDAC3 is recruited to the promoter of PPARγ2 by retinoic acid receptor-related orphan receptor-α and represses PPARγ2 expression by deacetylation, thereby downregulating the expression of its target fatty acid uptake genes and preventing hepatic steatosis [16]. In addition, HDAC3 is also co-recruited with prospero-related homeobox 1 protein to lipid metabolism genes by hepatocyte nuclear factor 4α and downregulates the expression of genes involved in TG synthesis and lipolysis [17]. A recent study also demonstrated that histone H3K4 methyltransferase MLL3/4 is recruited to the PPARγ responsible element of PPARγ2 and its target steatosis genes, stimulating their expression and resulting in hepatic steatosis [18]. Furthermore, our previous study demonstrated that the histone H3K9 demethylase JMJD2B stimulates the expression of PPARγ2 and its target genes, resulting in hepatic steatosis [15]. These reports have focused on the development of hepatic steatosis by the epigenetic regulation of fatty acid uptake-related genes such as *PPARγ2* and *CD36* by histone modification enzymes.

In addition to increased fatty acid uptake into hepatocytes, DNL also contributes significantly to TG accumulation in the pathogenesis of NAFLD [2]. Recently, it has been reported that the H3K9me2 demethylase, JMJD1C, is recruited to lipogenic promoter regions by USF-1 and demethylates H3K9me2, thus stimulating lipogenic gene expression [19]. This suggests that JMJD1C plays an important epigenetic regulatory role in the activation of lipogenesis. DNL occurs through the activation of SREBP1, ChREBP, and LXRα, which stimulate the expression of lipogenic genes, including *FAS*, *ACC*, and *SCD1* [2]. Among the lipogenic transcription factors, LXRα enhances SREBP1c expression by binding to the LXRE on its gene promoter, which subsequently induces the expression of lipogenic genes [3]. In addition, LXRα also stimulates the expression of lipogenic genes directly through binding to LXREs on their promoters. Thus, LXRα plays a critical role in lipogenesis [4]. The expression of LXRα and its downstream lipogenic genes are enhanced in liver biopsies from NAFLD patients. Thus, understanding the regulation of LXRα-dependent lipogenesis could provide an important therapeutic strategy against NAFLD.

Recently, the epigenetic regulation of LXRα-dependent lipogenesis by histone acetylation has been reported [20,21]. Histone deacetylase 5 (HDAC5) interacts with LXRα and inhibits its transcriptional activity by deacetylation at LXREs, leading to the reduction of LXRα-dependent lipogenesis [20]. HDAC5 expression level was reduced in HFD-fed obese mice, indicating that HDAC5 downregulation might contribute to the development of NAFLD. Furthermore, HDAC3 also downregulates the expression of LXRα in hepatic cells [21]. Accordingly, HDAC5 and HDAC3 can act as co-repressors of LXRα and are candidate therapeutic targets for the treatment of NAFLD. However, the epigenetic regulation of LXRα-dependent lipogenesis by histone methylation is poorly understood. In the current study, we investigated the epigenetic role of the histone demethylase JMJD2B in LXRα-dependent lipogenesis.

We observed that JMJD2B expression was enhanced in LXRα agonist-treated mice and HepG2 cells, as well as in HFD-fed mice and palmitate-treated HepG2 cells, concomitant with upregulated expression of LXRα-activated lipogenic genes, suggesting that JMJD2B is involved in LXRα-dependent lipogenesis. To explore whether JMJD2B participates in LXRα-dependent lipogenesis, we performed a loss-of-function study in HepG2 cells. Knockdown of JMJD2B using siRNA significantly reduced LXRα agonist-induced expression of LXRα and its target lipogenic genes, including *SREBF1*, *FAS*, *ACC*, and *SCD1*, suggesting that JMJD2B plays a role in LXRα-dependent lipogenesis. We then assessed the direct effects of JMJD2B on LXRα-mediated lipogenesis in HepG2 cells and mice. Adenovirus-mediated overexpression of JMJD2B in HepG2 cells enhanced the expression of LXRα and its target lipogenic genes, concomitant with increased intracellular TG accumulation. Furthermore, in vivo overexpression of JMJD2B using adenovirus also stimulated the expression of LXRα-dependent lipogenic genes and resulted in hepatic steatosis in HFD-fed mice. Taken together, these results indicate that JMJD2B stimulates LXRα-dependent lipogenesis, contributing to the development of hepatic steatosis.

JMJD2B is a histone demethylase responsible for converting the repressive histone marks H3K9me2 and H3K9me3 into H3K9me [8]. To characterize the mechanism by which JMJD2B stimulates LXRα-dependent lipogenesis, we investigated whether JMJD2B affects the enrichment of histone H3K9me2/3 near the LXRE on *SREBF1* promoter. To this end, we performed chromatin immunoprecipitation experiments in JMJD2B-overexpressing HepG2 cells. ChIP-PCR revealed that overexpression of JMJD2B increased both JMJD2B and LXRα enrichment at the LXRE of *SREBF1*, indicating that JMJD2B is recruited to the LXRE region together with LXRα. Consistently, overexpression of JMJD2B reduced H3K9me2 and H3K9me3 enrichment in the same location, indicating that JMJD2B removes the repressive histone marks H3K9me2 and H3K9me3 at LXREs of the *SREBF1* promoter. These results were consistent with those in LXRα agonist-treated HepG2 cells, which showed an increased enrichment of JMJD2B and LXRα in the vicinity of LXREs on the *SREBF1* promoter coupled with reduced levels of histone H3K9me2 and H3K9me3. These results suggest that the activation of LXRα induces the recruitment of JMJD2B to the LXRE of LXRα-dependent lipogenic genes, which subsequently leads to the removal of repressive histone marks H3K9me2 and H3K9me3 and results in the stimulation of LXRα-dependent lipogenesis.

Histone-modification enzymes including histone methyltransferases (or histone demethylases) and histone acetylases (histone deacetylases) are recruited to gene promoters by DNA-binding transcription factors or coactivators, and subsequently regulate gene expression through chromatin modification [22]. A previous study demonstrated that the H3K4 methyltransferase MLL1 was recruited to the LXRE on the *FAS* promoter by activating the signal cointegrator, and thereby upregulated *FAS* expression [22]. Thus, in the current study, we attempted to characterize how JMJD2B was recruited to LXRE. We speculated that JMJD2B was recruited to the LXRE region through a direct interaction with LXRE-bound LXRα. Therefore, we performed a co-immunoprecipitation assay in HepG2 cells infected with Ad-JMJD2B and incubated in the presence or absence of T0901378. Coimmunoprecipitation assay revealed an interaction between JMJD2B and LXRα in only both Ad-JMJD2B-infected and T09013178-treated HepG2 cells, suggesting that JMJD2B could be recruited to LXRα-activated lipogenic promoters by interacting with ligand-activated LXRα on LXRE, thus stimulating LXRα-dependent lipogenic genes by removing repressive H3K9me2/me3 marks around the LXRE.

The activation of LXRα is dependent on chromatin modifications, providing more insight into the regulation of LXRα-dependent gene expression. Activation of LXRα facilitates histone modifications at the LXRE of the LXRα target promoter genes by recruiting histone modification enzymes via several co-regulators, inducing chromatin remodeling [23]. LXRα significantly increases H3 and H4 acetylation, H3-S10 phosphorylation, and H3K4 methylation at the LXRE of *FAS* [23]. As mentioned previously, LXRα activation recruits H3K4 methyltransferases MLL1 to the LXRE of *FAS* and enhances H3K4 trimethylation (H3K4me3) [22], enhancing the expression of LXRα-dependent lipogenic genes. Our current data also suggest that LXRα activation recruits JMJD2B to the LXRE of *SREBF1* and reduces H3K9me2 and H3K9me3 marks at the LXRE of LXRα-dependent lipogenic genes. This suggests that LXRα activation induces JMJD2B-mediated removal of H3K9me2/me3 at LXREs and stimulates LXRα-dependent lipogenic genes.

Finally, we confirmed the inducible in vivo effects of JMJD2B on LXRα-dependent lipogenesis in mice infected with an Ad-JMJD2B. Overexpression of JMJD2B using recombinant Ad-JMJD2B induced hepatic TG accumulation, indicating that JMJD2B can promote the development of hepatic steatosis. The hepatic mRNA levels of *NR1H3* and its target lipogenic genes were significantly increased in Ad-JMJD2B-injected mice, consistent with the results in JMJD2B-overexpressed HepG2 cells. Taken together, these results suggest that JMJD2B stimulates LXRα-dependent lipogenesis, leading to the development of hepatic steatosis. Thus, JMJD2B could play an epigenetic regulatory role in LXRα-activated lipogenesis, and its overexpression could contribute to the development of hepatic steatosis.

## 4. Materials and Methods

### 4.1. Reagents

Dulbecco’s modified Eagle’s medium (DMEM), penicillin–streptomycin, and fetal bovine serum (FBS) were obtained from HyClone Laboratories Inc. (Logan, UT, USA). Antibodies against LXRα, H3K9me2, and H3K9me3 were purchased from Millipore (Billerica, MA, USA). Antibodies against JMJD2B, FAS, SREBP1c, and β-actin were purchased from Santa Cruz Biotechnology (Santa Cruz, CA, USA). Palmitate and T0901317 were purchased from Sigma–Aldrich (St. Louis, MO, USA).

### 4.2. Cell Culture and Treatment with T0901317 or Palmitate

The human hepatocellular carcinoma cell line, HepG2, was obtained from the American Type Culture Collection (Manassas, VA, USA). The HepG2 cells were cultured in DMEM supplemented with 10.0% heat-inactivated FBS and 100 U/mL penicillin–streptomycin at 37 °C in a humidified atmosphere with 5% CO_2_. Subsequently, the cells were seeded on plates and treated with 10 μM T0901317 and 100 μM palmitate for 24 h.

### 4.3. Triglyceride (TG) Measurement

TG levels from HepG2 cell lysates and liver tissues were determined as described previously [15].

### 4.4. Transfection of HepG2 Cells with siRNAs

To deplete JMJD2B, a duplex of siRNA targeting JMJD2B (sense: 5′-CCAGUUCAGU AUCAAUUAAAGCCCG-3′, antisense: 5′-CGGGCUUUAAUUGAUACUGAACUGGAG-3′) was designed and synthesized by Integrated DNA Technologies (Coralville, IA, USA). HepG2 cells were transfected with the siRNAs using Interferin^TM^ transfection reagent (Polyplus-Transfection Inc., New York, NY, USA).

### 4.5. Infection of HepG2 Cells with Recombinant Adenovirus

Adenovirus vectors encoding green fluorescent protein (GFP) or JMJD2B (Ad-GFP or Ad-JMJD2B) were purchased from Vector Biolabs (Malvern, PA, USA). HepG2 cells were infected with Ad-GFP or Ad-JMJD2B at a multiplicity of infection (MOI) ranging from 1 to 100 PFU/cell. Cells were washed and fresh medium was added. At 48 h post infection, the cells were harvested and analyzed.

### 4.6. Total RNA Preparation and Quantitative Real-Time Polymerase Chain Reaction (qPCR)

Total RNA was extracted from HepG2 cell lysates or liver tissues using TRIZOL^®^ reagent (Invitrogen, Carlsbad, CA, USA) according to the manufacturer’s instructions. cDNA was generated from 1 μg of total RNA using the GoScript™ Reverse Transcription System (Promega, Madison, WI, USA) in accordance with the manufacturer’s protocol. Quantitative real-time PCR was performed using a SYBR green premixed Taq reaction mixture with gene-specific primers. Gene-specific primers used in this study are listed in Appendix A.

### 4.7. Western Blot Analysis

Equal amounts of protein (20 μg/lane) from HepG2 cell lysates or liver tissues were resolved by 10% SDS-polyacrylamide gel electrophoresis (SDS-PAGE) and transferred onto polyvinylidene difluoride (PVDF) membranes (Millipore, Billerica, MA, USA). The membranes were blocked in 5% non-fat skim milk and probed with primary antibodies. After washing with Tween 20/Tris-buffered saline (T-TBS), membranes were incubated with a horseradish peroxidase-conjugated secondary antibody (1:1000) at room temperature for 1 h. Membranes were then washed three times with T-TBS, and proteins were detected using an enhanced chemiluminescence (ECL) Western Blot Detection Kit (Amersham, Uppsala, Sweden).

### 4.8. Chromatin Immunoprecipitation (ChIP)-qPCR

HepG2 cells were fixed with 1% formaldehyde at room temperature for 10 min. The crosslinked chromatin was sheared by sonication into 400 bp fragments using a Bioruptor Sonicator (Diagenode, Denville, NJ, USA). Samples were immunoprecipitated using 1–2 μg antibodies against JMJD2B, and LXRα, H3K9me2, and H3K9me3, or a nonspecific IgG control in the presence of secondary antibody conjugated to Dynabeads (Invitrogen, Carlsbad, CA, USA). Purified DNA was subjected to qPCR using the following primers: SREBP1c LXRE region sense: 5′-GTAAACGGAGGGTTGGAGC-3′, SREBP1c LXRE region antisense: 5′-CTGAATGGGGTTGGGGTTA-3′. ChIP data were normalized to those of the control IgG and expressed as a percentage of the input.

### 4.9. Luciferase Reporter Assay

The luciferase reporters, 3XLXRE-Luc, *SREBF1* promoter-Luc, and *FAS* promoter-Luc, were gifted by Prof. Lee (Seoul National University, Seoul, Korea). The human *SREBF1* promoter from −1564 to +1 relative to the transcription initiation site, and mouse *FAS* promoter from −1594 to +65 were inserted into pGL3 basic luciferase vector (Promega), respectively. The expression vectors for human JMJD2B and LXRα were obtained from Addgene (La Jolla, CA, USA). The transient transfection of reporters and expression vectors was performed using transfection reagents jetPRIME (Polyplus-Transfection Inc., New York, NY, USA). Luciferase activity was measured with a luminometer and normalized to that of galactosidase.

### 4.10. Co-Immunoprecipitation (CoIP) Assay

HepG2 cells were infected with adenovirus expressing JMJD2B (Ad-JMJD2B) and then incubated in the presence or absence of T0901317 for 24 h. The cells were harvested and lysed in lysis buffer (25 mM Tris-HCL pH7.4, 150 mM NaCl, 1% NP-40, 1 mM EDTA, 5% glycerol) containing 0.5 mM PMSF and 1 × protease inhibitor (TransLab Biosciences, Daejon, Korea) for 30 min on ice. The lysates were then centrifuged at 13,000 rpm for 30 min at 4 °C. Amounts of 500 μg protein determined using the Bradford assay. Samples were incubated with antibody against JMJD2B (Santa Cruz, CA, USA) or nonspecific IgG control for overnight at 4 °C via gentle rocking. Protein A/G Plus-Agarose beads (Santa Cruz, CA, USA) were added to capture the immune complexes (protein antibody beads) via mixing for 1 h at 4 °C on rotator. Beads were washed three times in wash buffer (25 mM Tris-HCL pH 7.4, 150 mM NaCl, 1% NP-40, 1 mM EDTA, 5% glycerol) with 5 min rotation in between. The proteins were eluted from the beads with 1× SDS loading buffer at 95 °C for 5 min. Co-immunoprecipitated proteins were analyzed by immunoblotting with LXRα antibody.

### 4.11. Animal Experiments

To make a mouse model of fatty liver disease, C57BL/6J mice (8 weeks of age) were treated with T0901317 daily by oral gavage for 5 days or were fed an HFD for 12 weeks. Furthermore, to overexpress JMJD2B in mice, C57BL/6 mice (8 weeks of age) were injected with a total of 1 × 109 PFU recombinant adenovirus (Ad-GFP or Ad-JMJD2B) via tail vein injection. After injection, adenovirus-injected mice were fed an HFD for 2 weeks. At the end of the treatment period, the mice were sacrificed and the liver was immediately removed and frozen at −80 °C. All animal experiments were approved by Pusan National University Institutional Animal Care and Use Committee in accordance with the established ethical and scientific care procedures (PNU-2017-1483).

### 4.12. Statistical Analysis

The data shown in this study are expressed as mean ± SD. The data were analyzed using one-way ANOVA, and the differences between means were determined using the Tukey–Kramer post-hoc test. Values were considered statistically significant at *p* < 0.05.

## 5. Conclusions

Based on the current data, we propose an epigenetic role of JMJD2B in LXRα-mediated stimulation of lipogenesis (Figure 10). LXRα activation induces the recruitment of JMJD2B to the LXRE on the promoters of LXRα target lipogenic genes where it then removes H3K9me2 and H3K9me3 marks at the LXRE, enhancing lipogenic gene expression. The current study provides a novel mechanism for the regulation of LXRα-dependent lipogenic gene transcription.

## Figures and Tables

**Figure 1 ijms-21-08313-f001:**
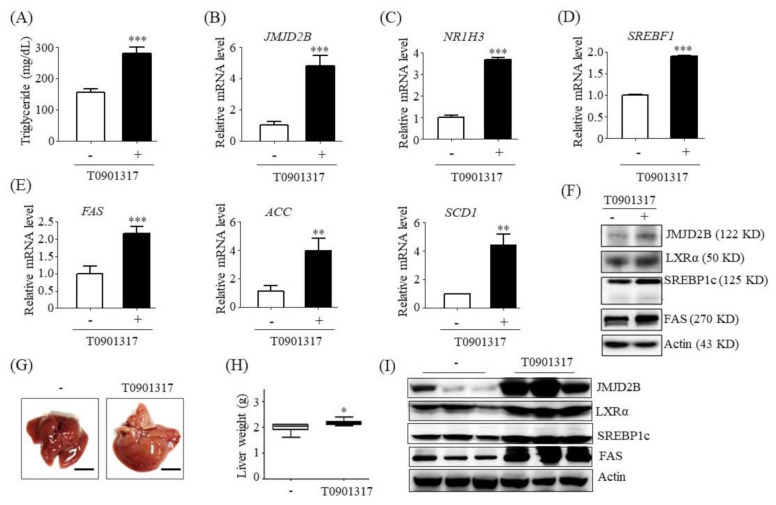
Jumonji domain-containing protein 2B (JMJD2B) expression level was upregulated in liver X receptor α (LXRα) agonist-treated HepG2 cells and the liver of LXRα agonist-treated mice. (**A**–**F**) HepG2 cells were incubated with T0901317 for 24 h. (**A**) Intracellular TG were measured using a triglyceride (TG) assay kit. (**B**–**E**) mRNA levels of *JMJD2B*, *NR1H3*, *SREBF1*, fatty acid synthase *(FAS*), acetyl CoA carboxylase (*ACC*), and stearoyl-CoA desaturase 1 (*SCD1*) were measured by qPCR. Data are presented as means ± SD from three independent experiments. ** *p* < 0.01, *** *p* < 0.001 vs. untreated control. (**F**) Protein expression levels of JMJD2B, LXRα, SREBP1c, and FAS were measured by western blotting. The SREBP1c shown here is precursor form (125 KD). (**G**–**I**) C57BL/6J mice were administrated with T0901317 by oral gavage daily for 5 days. (**G**) Representative liver photographs. (scale bar = 1 cm) (**H**) Liver weight. (**I**) Protein levels of JMJD2B, LXRα, SREBP1c, and FAS were measured by western blotting. The SREBP1c shown here is mature form (68 KD). Densitometric analysis of band intensity is shown in Appendix A. Data are presented as means ± SD from 6 mice (*n* = 6). * *p* < 0.05 vs. untreated control mice.

**Figure 2 ijms-21-08313-f002:**
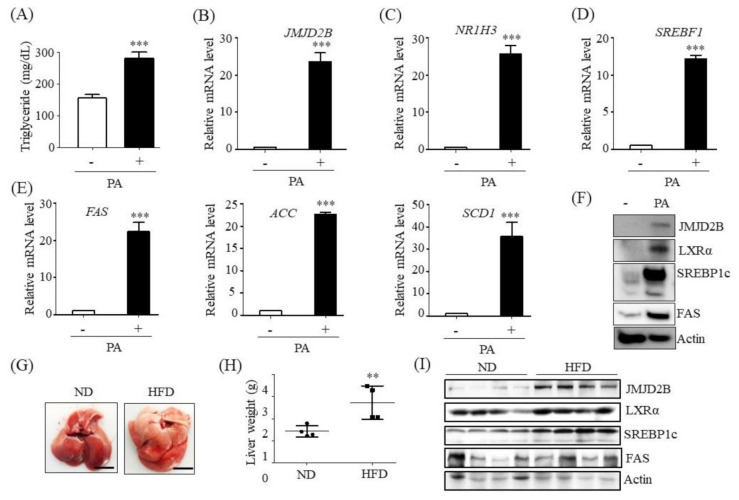
JMJD2B expression level was upregulated in palmitate (PA)-treated HepG2 cells and the liver of HFD-fed mice. (**A**–**F**) HepG2 cells were incubated with PA for 24 h. (**A**) Intracellular TG were measured using a TG assay kit. (**B**–**E**) mRNA levels of *JMJD2B*, *NR1H3*, *SREBF1*, *FAS*, *ACC*, and *SCD1* were measured by qPCR. Data are presented as means ± SD from three independent experiments. *** *p* < 0.001 vs. untreated control. (**F**) Protein expression levels of JMJD2B, LXRα, SREBP1c, and FAS were measured by western blotting. The SREBP1c shown here is precursor form (125 KD). (**G**–**I**) C57BL/6J mice were fed a high-fat diet (HFD) for 12 weeks. (**G**) Representative liver photographs. (scale bar = 1 cm) (**H**) Liver weight. (**I**) Protein levels of JMJD2B, LXRα, SREBP1c, and FAS were measured by western blotting. The SREBP1c shown here is mature form (68 KD). Densitometric analysis of band intensity is shown in Appendix A. Data are presented as means ± SD from 6 mice (*n* = 6). ** *p* < 0.01 vs. normal diet (ND) mice.

**Figure 3 ijms-21-08313-f003:**
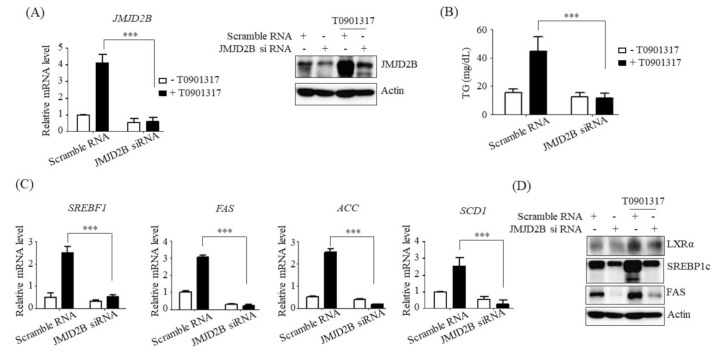
Knockdown of JMJD2B prevented LXRα agonist-induced lipogenic gene expression in HepG2 cells. HepG2 cells were transfected with scramble RNA or JMJD2B siRNA and then treated with T0901317. (**A**) JMJD2B expression was determined by qPCR and western blotting. (**B**) Intracellular TG levels were measured using a TG assay kit. (**C**) mRNA levels of *SREBF1*, *FAS*, *ACC*, and *SCD1* were determined by qPCR. (**D**) Protein levels of LXRα, SREBP1c, and FAS were measured by western blotting. The SREBP1c shown here is precursor form (125 KD). Data are presented as means ± SD from three independent experiments. *** *p* < 0.001 vs. scramble siRNA.

**Figure 4 ijms-21-08313-f004:**
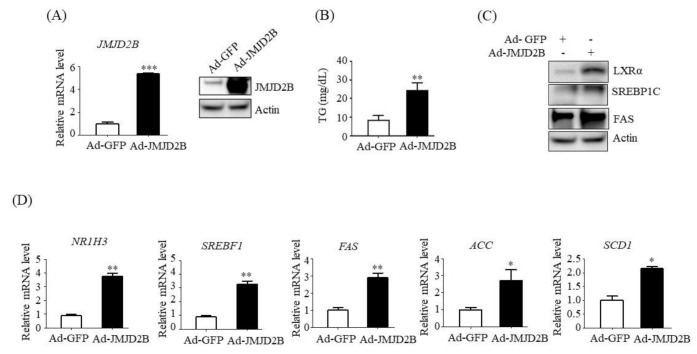
Overexpression of JMJD2B stimulated LXRα-dependent lipogenic genes in HepG2 cells. HepG2 cells were infected with Ad-GFP or an adenovirus containing JMJD2B (Ad-JMJD2B). (**A**) JMJD2B expression was measured using qPCR and western blotting. (**B**) Intracellular TG levels were measured using a TG assay kit. (**C**) Protein levels of LXRα, SREBP1c, and FAS were measured by western blotting. The SREBP1c shown here is precursor form (125 KD). (**D**) mRNA levels of *NR1H3*, *SREBF1*, *FAS*, *ACC*, and *SCD1* were measured by qPCR. Data are presented as means ± SD from three independent experiments. * *p* < 0.05, ** *p* < 0.01, *** *p* < 0.001 vs. control Ad-GFP.

**Figure 5 ijms-21-08313-f005:**
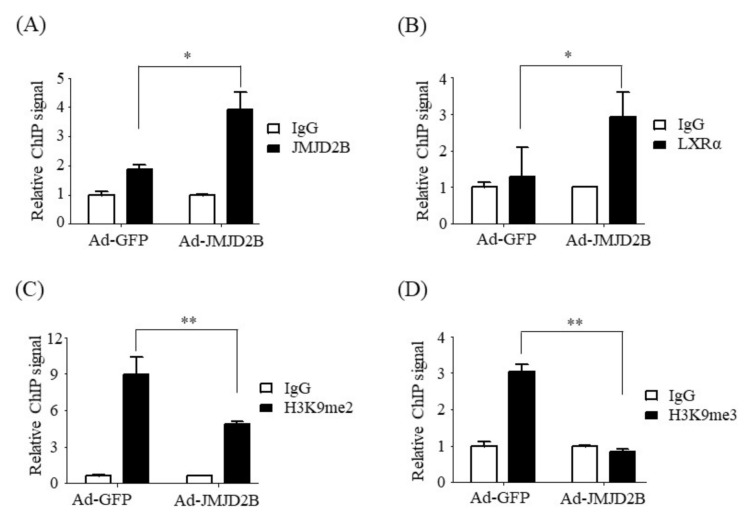
JMJD2B reduced the enrichment of H3K9me2 and H3K9me3 at the LXR response element (LXRE) on the *SREBF1* promoter. HepG2 cells were infected with adenovirus Ad-GFP or Ad-JMJD2B. (**A**,**B**) The recruitment of JMJD2B (**A**) and LXRα (**B**) to the LXRE on the *SREBF1* promoter was analyzed by chromatin immunoprecipitation (ChIP)-qPCR. (**C**,**D**) The enrichment of H3K9me2 (**C**) and H3K9me3 (**D**) at the LXRE on the *SREBF1* promoter was analyzed by ChIP-qPCR. Data are presented as the mean ± SD from three independent experiments. * *p* < 0.05, ** *p* < 0.01 vs. control Ad-GFP.

**Figure 6 ijms-21-08313-f006:**
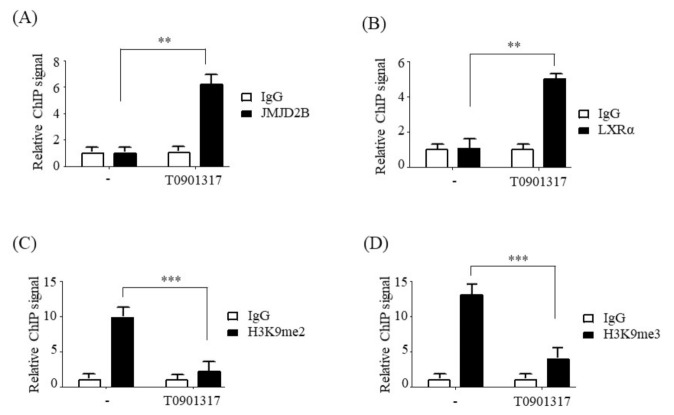
LXRα activation reduced the enrichment of H3K9me2 and H3K9me3 at the LXRE on the *SREBF1* promoter. HepG2 cells were incubated with T0901378 for 24 h. (**A**,**B**) The recruitment of JMJD2B (**A**) and LXRα (**B**) to the LXRE on the *SREBF1* promoter was analyzed by ChIP-qPCR. (**C**,**D**) The enrichment of H3K9me2 (**C**) and H3K9me3 (**D**) in the vicinity of the LXRE on the *SREBF1* promoter was analyzed by ChIP-qPCR. Data are presented as the mean ± SD from three independent experiments. ** *p* < 0.01, *** *p* < 0.001 vs. untreated control.

**Figure 7 ijms-21-08313-f007:**
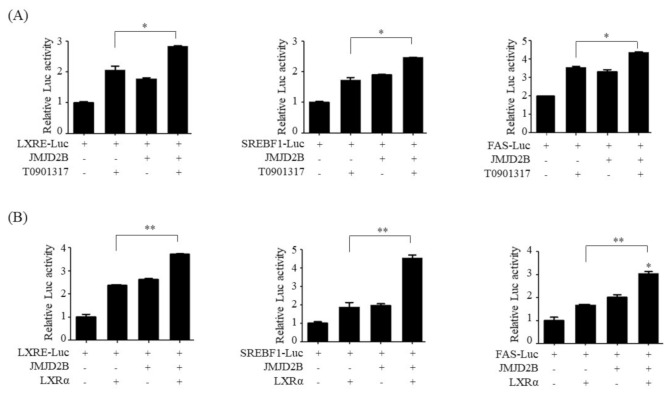
JMJD2B stimulated LXRα-activated transcriptional activity of LXRE-containing luciferase reporters. (**A**) HepG2 cells were transfected with Luc-reporters including 3xLXRE-Luc, *SREBF1*-Luc and *FAS*-Luc, and JMJD2B expression vector (pCMV-JMJD2B), then incubated with T0901317 for 24 h. Luciferase activities were determined using a kit. Data are presented as the mean ± SD from three independent experiments. ** p* < 0.05 vs. only T0901317 treatment. (**B**) HepG2 cells were transfected with Luc-reporters, JMJD2B expression vector (pCMV-JMJD2B), and LXRα expression vector (pCMV-LXRα). Luciferase activities were determined using a kit. ** *p* < 0.01 vs. only pCMV-LXRα.

**Figure 8 ijms-21-08313-f008:**
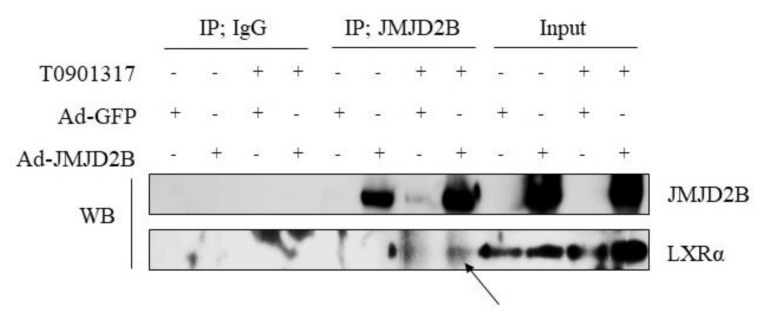
JMJD2B interacted with activated LXRα. HepG2 cells were infected with Ad-GFP or Ad-JMJD2B and then incubated in the presence or absence of T09013178 for 24 h. Protein extracts were immunoprecipitated with JMJD2B antibody-agarose beads, and the interaction was detected with immunoblot analysis using LXRα antibody. Arrow indicates LXRα band detected in JMJD2B antibody-mediated immunoprecipitants. The alternative data is shown in Appendix A where intracellular JMJD2B.

**Figure 9 ijms-21-08313-f009:**
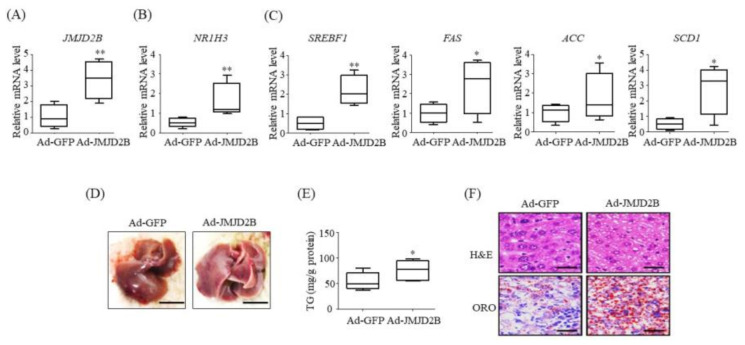
Overexpression of adenovirus-mediated JMJD2B stimulated hepatic LXRα-dependent lipogenic genes. C57BL/6 mice (8-week-old) were injected with adenovirus Ad-GFP or Ad-JMJD2B. After injection, Ad-injected mice were fed HFD for 2 weeks. (**A**–**C**) mRNA levels of *JMJD2B* (**A**), *NR1H3* (**B**), and LXRα-target lipogenic genes *SREBF1*, *FAS*, *ACC*, and *SCD1* (**C**) were determined by qPCR. (**D**) Representative photographs (scale bar = 1 cm). (**E**) Hepatic TG levels were measured by TG assay kit. (**F**) H&E and ORO staining (scale bar = 50 μm). Data are presented as the mean ± SD from 6 mice (*n* = 6). * *p* < 0.05, ** *p* < 0.01 vs. Ad-GFP-injected mice.

**Figure 10 ijms-21-08313-f010:**
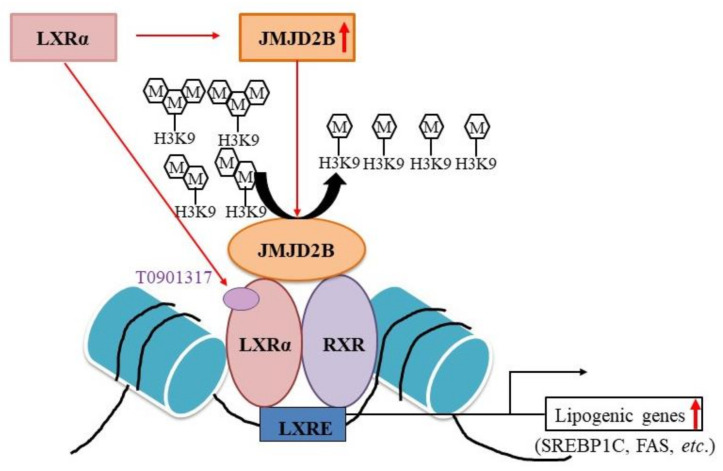
Proposed epigenetic role of JMJD2B in LXRα-mediated stimulation of lipogenesis.

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
