# Peer review of "Histone H3K9 Demethylase JMJD2B Plays a Role in LXRα-Dependent Lipogenesis"

_ijms, 2020, doi:10.3390/ijms21218313_

Round 1
Reviewer 1 Report
The paper by Kim and collaborators is an interesting documentation of the role of JMJD2B in the control of liver lipogenesis through LXRalpha. They found that JMJD2D expression was upregulated in HepG2 cells and the livers of mice treated with the LXRα agonist (T09), and in palmitate (PA)-treated HepG2 cells and the liver of HFD-fed mice. Of note, Knockdown of JMJD2D prevented LXRα-dependent lipogenesis in HepG2 cells, and overexpression of JMJD2B induced LXRα-dependent lipogenesis (still in HepG2 cells). Overexpression of JMJD2B in HepG2 cells reduced the enrichment of H3K9me2 and H3K9me3 at LXREs on the SREBF1 promoter via the recruitment of JMJD2B and LXRα. It was also reported that JMJD2B stimulated LXRα-activated transcriptional activity. The authors tried to demonstrate that JMJD2B interacted directly with activated LXRα. Further, adenovirus-mediated JMJD2B overexpression stimulated LXRα-dependent lipogenesis and induced hepatic steatosis in WT mice.
In general, it is a rather well-conducted study ; the rational and the specific hypotheses underlying this study are rather globally clear. However, I don’t find that all the results conclusive and therefore I have a number of concerns and questions to rise.
1- Lane 94 : Please, provide literature supporting that activated LXRalpha promotes its own expression in human liver cells (such as HepG2).
2- Western-blot Fig. 1F are not conclusive, at least for JMJD2B and LXRalpha. Please, provide here the respective Molecular Weight for each protein. Moreover, proper quantification against actin is required (idem figure 2I and 4C).
Did you also test another LXR agonist such as GW3965 ? If so, what are the results ?
3- Which SREBP1c are you studying here. The precursor or the mature form ? Please, clarify.
4- Lane 114 : Please, recall to the readers why you used palmitate in the following experiment set-up. Is it to mimic somehow HFD ?
5- Figure 3D : Western-blot for LXRalpha has to be done again since it is far to be useful and conclusive in the present form. Perhaps could you use nucleus extracts instead of whole cell lysates ?
6- Figure 4c :Why there is no dectection of LXRalpha protein in “Ad-GFP condition“, while you get a clear signal for the same condition with Input (see Figure 8, Input, lane Ad-GFP).
7- To show that indeed the LXRE of the SREBF1 promoter gene is bound by a protein complex that contains the heterodimer LXRalpha-JMJD2B, the best would be to do “Re-Chip“.
8- In an attempt to show that LXRalpha directly interacts with JMJD2B, a co-immunoprecipiation was peformed, using as a precipitating Ab, the JMJD2B antibody (figure 8). However, it does not take into account the genomic DNA part since authors worked here with soluble proteins only.
Because the data here again are not fully clear cut and conclusive, the reviewer would like to see the opposite, i.e., precipitate with anti-LXRalpha antibody and then reveal by WB with the anti-JMJD2B antibody. Perhaps the use of Tagged proteins (HA, C-myc, GFP,…) for LXR and JMJD2B could be useful here if antibodies for LXRa and JMJD2B are difficult to work with.
9- In general, siRNA against JMJD2B was used to knock-down its expression. Could you reinforce your finding using a pharmacological inhibitor for JMJD2B, if available ?
10- Do you also get data - similar to Figure 5 and Figure 6 - with Palmitic Acid instead of T0901317? Do they copy those with T0901317 ? Please provide them in supplementary data.
11- You claimed that “hepatic TG levels were significantly higher in Ad-JMJD2B-injected mice than in Ad-GFP-injected mice“, yet no statistical data are provided that support this notion. Please, verify this point.
12- It is mandatory to have data similar to those for Figure 5 and figure 6 (HepG2 cells) with liver samples of Figure 1 I and Figure 9 to definitively close the story since HepG2 cells are of human origin and Figure 1 I and Figure 9 are data with mouse.
13- In the discussion section, it would be nice to open the discussion to other cell types such as the macrophages in which LXR(a) also plays a role in both cholesterol efflux and inflammation. What about the connexion between LXRa and JMJD2B in this context ?
Other points :
1) Lane 82 : suggesting “that“
2) 4.9. Luciferase reporter assay : “SREBF1 promoter-Luc, and FAS promoter-Luc“ : do you mean that you worked with a LXRE within the original context of the natural promoter (several hundreds bp), respectively SREBF1 and FAS ? and not with a single copy of LXRE from either SREBF1 or FAS genes cloned in front of a TK-Luc (or SV40-Luc) rapporter vector ? This is not fully clear for me.
3) Figure 9F is too small ; please make it bigger.
Reviewer 2 Report
In this manuscript, the authors explored an epigenetic role of JMJD2B in LXRα-mediated stimulation of lipogenesis by removing the repressive histone marks (H3K9me2 and H3K9me3), thus contributing to the development hepatic steatosis. The manuscript was well written with proper design and tons of data. I listed some general comments on this manuscript.
In Fig. 1B, how LXRα agonist T0901317 leads to JMJD2B expression?
In Fig. 3A and 4B, JMJD2B is clearly expressed in control cells, but JMJD2B is not detected in the input (control) of Fig. 8. This is a serious issue and need to be addressed.
The authors only analyzed the genes related with lipogenesis, how is the change of lipolysis and the expression of lipolysis-related genes by JMJD2B overexpression?
It is worth to try the effect of JMJD2B overexpression or knockdown on the genes other than LXRa-target genes.
Author Response
1. In Fig. 1B, how LXRα agonist T0901317 leads to JMJD2B expression?
Response;
- Activated LXRα by T0901317 binds to LXRE on the promoter of JMJD2B and stimulates JMJD2B expression.
2. In Fig. 3A and 4B, JMJD2B is clearly expressed in control cells, but JMJD2B is not detected in the input (control) of Fig. 8. This is a serious issue and need to be addressed.
Response;
- We appreciate your critical pointing. If longer exposure, intracellular JMJD2B should be appear in the input of Fig. 8.
3. The authors only analyzed the genes related with lipogenesis, how is the change of lipolysis and the expression of lipolysis-related genes by JMJD2B overexpression?
Response;
- We appreciate your comments. We did not examined the expression of lipolysis genes. However, as you suggested, we will check the expression of lipolysis-related genes in JMJD2B overexpressed cells later.
4. It is worth to try the effect of JMJD2B overexpression or knockdown on the genes other than LXRa-target genes.
Response;
- We appreciate your valuable comment. Later, we will investigate the effects of JMJD2B on the genes other than LXRa-target genes.
Reviewer 3 Report
Final comments:
This paper shows histone H3K9 demethylase JMJD2B upregulates lipogenesis of hepatocyte by acitivation of LXRa and initiates NASH. Their data looks very clear and interesting. These data prompt us that LXR-a will be a good target for treating NASH. If the authors have any idea how to inhibit lipogenesis of liver please add discussions.
I think this paper is good for publication in this present form, discussions about clinical applications will attract clinical hepatologists.
Author Response
This paper shows histone H3K9 demethylase JMJD2B upregulates lipogenesis of hepatocyte by acitivation of LXRa and initiates NASH. Their data looks very clear and interesting. These data prompt us that LXR-a will be a good target for treating NASH. If the authors have any idea how to inhibit lipogenesis of liver please add discussions.
I think this paper is good for publication in this present form, discussions about clinical applications will attract clinical hepatologists.
Response;
- We appreciate your comment. LXRalpha is highly expressed in the livers of patients with NAFLD, increasing with the severity of NASH. As the current study suggest that JMJD2B acts as coactivator for LXR-mediated lipogenesis, the expression of JMJD2B need to be further verified in the livers of patients with NAFLD.
Furthermore, since LXRalpha is a promising target for drug development of NAFLD, JMJD2B might be also a good therapeutic target epigenectically against NAFLD.
Round 2
Reviewer 1 Report
The authors have tried to do their best within the period of revision (10 days). Yet, I still have the following points/issues to fix before publication.
1- You claimed that you provided western blot data of Fig.1F with molecular weight for each protein. However, in the revised version 2 of your MS, it is not the case ! Please, correction is here required.
2- - You claimed that you quantified the band density of western blots in Figure 1I, and 2I, and represented the bar graph in supplementary figure (Figure S1). Again, in the revised version 2 of your MS, it is not the case since only primer sequences for PCR are present in the file for Supplementary Data ! Please, correction is here required.
3- You claimed that “The SREBP1c form shown in western blotting using liver extracts (Fig. 1I and Fig. 2I) is mature form (68 KD). The other SREBP1c form (Fig. 1F, Fig. 2F, Fig. 3D, Fig. 4C) is precursor form (125 KD)“. Please, indicate for the reader these informations in the revised version 2 of your MS !!
4- Please, recall to the readers why you used palmitate in the following experiment set-up. Is it to mimic somehow HFD ? You answered to me BUT not to the readers of your MS…..therefore, do it properly.
5- Figure 4c : Why there is no detection of LXRalpha protein in “Ad-GFP condition“, while you get a clear signal for the same condition with Input (see Figure 8, Input, lane Ad-GFP).
Your Response;
- We appreciate your critical pointing. If longer exposure, intracellular LXRalpha protein band should be appeared in Ad-GFP condition of Figure 4C. SO PLEASE, GIVE THE READER THIS LONGER EXPOSURE TO MAKE SURE THAT YOU CAN DETECT LXRalpha IN Ad-GFP condition, instead of the actual picture.
6- It would make sense to add the western-blot with IP LXRalpha (WB with JMJD2B antibody) in the supplementary data section.
7- (point 4.9 of my initial review) : either you used ISOLATED LXRE from FAS and SREBP cloned in front of a SV40 promoter OR yu used the ORIGINAL NATURAL promoters of SREBP and FAS (that contain their own TATA box and LXRE), but not the ORIGINAL promoter fused to SV40 LUC…….otherwise, you do have two promoters, two TATA Box in your constructs !!!…… Please, make sure of what you used here.
Author Response
1- You claimed that you provided western blot data of Fig.1F with molecular weight for each protein. However, in the revised version 2 of your MS, it is not the case ! Please, correction is here required.
Response;
We have provided molecular weight for each protein in Fig. 1F.
2- You claimed that you quantified the band density of western blots in Figure 1I, and 2I, and represented the bar graph in supplementary figure (Figure S1). Again, in the revised version 2 of your MS, it is not the case since only primer sequences for PCR are present in the file for Supplementary Data ! Please, correction is here required.
Response;
We have provided the bar graph in the supplementary Fig. S1.
3- You claimed that “The SREBP1c form shown in western blotting using liver extracts (Fig. 1I and Fig. 2I) is mature form (68 KD). The other SREBP1c form (Fig. 1F, Fig. 2F, Fig. 3D, Fig. 4C) is precursor form (125 KD)“. Please, indicate for the reader these informations in the revised version 2 of your MS !!
Response;
As your suggestion, we have described the SREBP1c form in the figure legends. (red font)
4- Please, recall to the readers why you used palmitate in the following experiment set-up. Is it to mimic somehow HFD ? You answered to me BUT not to the readers of your MS…..therefore, do it properly.
Response;
As your suggestion, we have described the reason why palmitate was used in the manuscript. (red font)
5- Figure 4c : Why there is no detection of LXRalpha protein in “Ad-GFP condition“, while you get a clear signal for the same condition with Input (see Figure 8, Input, lane Ad-GFP).
Your Response;
We appreciate your critical pointing. If longer exposure, intracellular LXRalpha protein band should be appeared in Ad-GFP condition of Figure 4C. SO PLEASE, GIVE THE READER THIS LONGER EXPOSURE TO MAKE SURE THAT YOU CAN DETECT LXRalpha IN Ad-GFP condition, instead of the actual picture.
Response;
To confirm the expression of LXRα in Figure 4C, we repeated the western blotting under same conditions. As shown in newFig. 4C, we could see the intracellular LXRα in Ad-GFP condition. Therefore, we replaced the new LXRα data in Fig. 4C.
6- It would make sense to add the western-blot with IP LXRalpha (WB with JMJD2B antibody) in the supplementary data section.
Response;
As your suggestion, we have added the data in the supplementary Fig. S2.
7- (point 4.9 of my initial review) : either you used ISOLATED LXRE from FAS and SREBP cloned in front of a SV40 promoter OR yu used the ORIGINAL NATURAL promoters of SREBP and FAS (that contain their own TATA box and LXRE), but not the ORIGINAL promoter fused to SV40 LUC…….otherwise, you do have two promoters, two TATA Box in your constructs !!!…… Please, make sure of what you used here.
Response;
We made a mistake. The human SREBF1 promoter from -1564 to +1 relative to transcription initiation site, and mouse FAS promoter from -1594 to +65 were inserted into pGL3 basic luciferase vector (Promega), respectively. We have described them in the Materials and Methods. (red font)
